# Formoterol Acting via β2-Adrenoreceptor Restores Mitochondrial Dysfunction Caused by Parkinson’s Disease-Related UQCRC1 Mutation and Improves Mitochondrial Homeostasis Including Dynamic and Transport

**DOI:** 10.3390/biology13040231

**Published:** 2024-03-30

**Authors:** Jui-Chih Chang, Huei-Shin Chang, Yi-Chun Chao, Ching-Shan Huang, Chin-Hsien Lin, Zhong-Sheng Wu, Hui-Ju Chang, Chin-San Liu, Chieh-Sen Chuang

**Affiliations:** 1Center of Regenerative Medicine and Tissue Repair, Institute of ATP, Changhua Christian Hospital, Changhua 500, Taiwan; 2Inflammation Research & Drug Development Center, Changhua Christian Hospital, Changhua 500, Taiwan; 3Department of Neurology, National Taiwan University Hospital, Taipei 100, Taiwan; 4Department of General Research Laboratory of Research, Changhua Christian Hospital, Changhua 500, Taiwan; 5Department of Neurology, Changhua Christian Hospital, Changhua 500, Taiwan; 6Vascular and Genomic Center, Institute of ATP, Changhua Christian Hospital, Changhua 500, Taiwan; 7Graduate Institute of Integrated Medicine, College of Chinese Medicine, China Medical University, Taichung 404, Taiwan; 8College of Medicine, National Chung Hsing University, Taichung 402, Taiwan

**Keywords:** formoterol, Parkinson’s disease, ubiquinol-cytochrome c reductase core protein 1, mitochondrial function, mitochondrial dynamics

## Abstract

**Simple Summary:**

Formoterol, an FDA-approved long-acting beta2-adrenergic receptor agonist, has shown potential benefits in various diseases, yet its effectiveness in Parkinson’s disease (PD) is uncertain. The lack of a comprehensive understanding of formoterol’s mechanism, particularly in mitochondrial remodeling, contributes to this uncertainty. PD involves mitochondrial dysfunction, leading to disruptions in energy production, heightened oxidative stress, and impaired mitochondrial dynamics and turnover. Understanding this link is vital for developing targeted therapies for PD. To explore this, this study used a cell model that mimics a specific genetic form of PD. This model displayed the above problems with mitochondria. The study found that after treating these cells with formoterol, there were positive effects. The medication enhanced the growth and survival of PD-associate mutant cells while protecting against stress. Crucially, it contributed to restoring normal mitochondrial function and machinery, promoting a rebalance in mitochondrial dynamics, including changes in morphology (fusion/fission), movement, and transport. This effect was achieved by influencing specific cell signals and proteins associated with mitochondrial health. This study underscores formoterol’s pivotal role as a mitochondrial dynamic balance regulator, positioning it as a promising therapeutic candidate for PD.

**Abstract:**

Formoterol, a β2-adrenergic receptor (β2AR) agonist, shows promise in various diseases, but its effectiveness in Parkinson’s disease (PD) is debated, with unclear regulation of mitochondrial homeostasis. This study employed a cell model featuring mitochondrial ubiquinol-cytochrome c reductase core protein 1 (UQCRC1) variants associated with familial parkinsonism, demonstrating mitochondrial dysfunction and dynamic imbalance, exploring the therapeutic effects and underlying mechanisms of formoterol. Results revealed that 24-h formoterol treatment enhanced cell proliferation, viability, and neuroprotection against oxidative stress. Mitochondrial function, encompassing DNA copy number, repatriation, and complex III-linked respiration, was comprehensively restored, along with the dynamic rebalance of fusion/fission events. Formoterol reduced extensive hypertubulation, in contrast to mitophagy, by significantly upregulating protein Drp-1, in contrast to fusion protein Mfn2, mitophagy-related protein Parkin. The upstream mechanism involved the restoration of ERK signaling and the inhibition of Akt overactivity, contingent on the activation of β2-adrenergic receptors. Formoterol additionally aided in segregating healthy mitochondria for distribution and transport, therefore normalizing mitochondrial arrangement in mutant cells. This study provides preliminary evidence that formoterol offers neuroprotection, acting as a mitochondrial dynamic balance regulator, making it a promising therapeutic candidate for PD.

## 1. Introduction

Formoterol is a long-acting beta2-adrenergic receptor (β2AR) agonist and a type of medication that is used for treating asthma and chronic obstructive pulmonary disease (COPD), as it relaxes the muscles of the airways, thus affording easier breathing [1]. It is also often used in combination with other medications to help control the symptoms of Parkinson’s disease (PD), such as tremors, stiffness, and difficulty with movement [2]. The relationship between β2AR agonists and the risk of PD remains a subject of debate. Although some studies have found no significant association between β2AR medications and PD risk [3,4], others have suggested that the chronic use of β2AR blockers has neuroprotective effects [5] and reduces the risk of PD [6,7,8]. Moreover, their underlying mechanism has been suggested to be related to the regulation of α-synuclein transcription [9], which increases the activity of dopamine, a neurotransmitter, and decreases neuroinflammation in the brain [10]. Furthermore, formoterol stimulates the production of signaling molecules (such as cyclic AMP) that play a key role in many physiological processes, including the regulation of energy metabolism [11]. Recently, the effect of formoterol on mitochondria, the “powerhouses” of cells, was noted, which has been consistently correlated with the restoration of mitochondrial activity, biogenesis, and homeostasis in different cells and tissues, thus potentially contributing to its effects on metabolism and energy production [12,13]. However, the precise molecular mechanisms by which formoterol affects mitochondrial function, especially in regulating mitochondrial dynamics in PD, remain unknown.

Mitochondria are dynamic organelles that undergo continuous processes of fusion and fission. This phenomenon maintains the hemostasis of mitochondria and the integrity of mitochondrial DNA (mtDNA) in many ways. Mitochondrial fusion shares materials to sustain stress-induced damage. In contrast, mitochondrial fission helps to distribute mitochondria evenly within a cell through mitochondrial transport, together with microtubules, to ensure that all parts of the cell have access to the energy required by them [14]. This process can trigger cells to discard damaged mitochondria via mitophagy to control mitochondrial quality [15]. Notably, mitochondrial dynamic dysfunction also implicates the regulation of cell signaling, the cell cycle, and apoptotic mechanisms in stresses caused by genetic and environmental factors and beyond to cellular energy support [16]. Thus, disruptions in the balance of mitochondrial dynamics have been implicated in the development and progression of various diseases, including PD.

Mitochondrial ubiquinol-cytochrome c reductase core protein 1 (UQCRC1)-associated parkinsonism with polyneuropathy is a rare genetic disorder that is caused by a missense mutation, c.941A > C (p.Tyr314Ser) in the UQCRC1 gene [17,18]. The exact underlying mechanisms involve energy deficits, oxidative stress, and loss of the engagement of cytochrome c to trigger neuronal death [17,19,20]; however, the impact on mitochondrial dynamics remains unclear. In particular, a marked dysregulation of mitochondrial dynamics, as revealed by abnormally elongated mitochondria and an irregular shape, is consistently observed in both UQCRC1-mutant neurons and animals [17,19]. Therefore, in the present study, through the exploration of the machinery related to the maintenance of mitochondrial homeostasis (including mitochondrial fission, fusion, transport, and biogenesis) in a cellular model of UQCRC1 parkinsonism, the therapeutic potential of the targeting of mitochondrial dynamics by formoterol for the future therapy of PD was examined.

## 2. Materials and Methods

### 2.1. Cell Culture and Treatment

The wild-type (WT) and mutant *UQCRC1* knock-in human neuroblastoma SH-SY5Y cell lines were gifts from the laboratory of Dr Chin-Hsien Lin [18]. The cells were individually transfected with CRISPR/Cas9 plasmids without site mutation (as a control in this experiment, WT) or carried the c.941A > C (p.Tyr314Ser) heterozygous variant of the *UQCRC1* gene identified in patients with PD (*UQCRC1* mutation) and confirmed by PCR amplification and Sanger sequencing for confirmation [17]. All cells were cultured in DMEM/F12 (Life Technologies, GIBCO BRL, Rockville, MD, USA) supplemented with 10% FBS (BIOSER, Buenos Aires, Argentina), 100 U/mL penicillin, and 100 µg/mL streptomycin (Life Technologies, GIBCO BRL), then grown in a humidified atmosphere containing 5% CO_2_.

Mutant cells were treated with 1 μM formoterol alone or in combination with 100 μM propranolol, a β2AR antagonist, for 24 h in the presence or absence of 10 μM tert-butylhydroperoxide (tBH)-induced oxidative stress. The effective doses of various compounds were determined following preliminary WST-1 analysis, with none observed to adversely affect WT cell viability (see Appendix A).

### 2.2. GFP-Labeled Mitochondria

Cells at 80% confluency were transfected with 40 μg of plasmid DNA encoding mitochondrial-matrix-localized AcGFP (import targeting sequence of cytochrome c oxidase subunit 8, COX8) (Clontech, Palo Alto, CA, USA) using electroporation (ECM; BTX Harvard Apparatus, Holliston, MA, USA) according to our previous study [21]. Thirty-six hours later, the cells were transferred to a normal growth medium for 48 h, followed by G418 selection (500 mg/mL).

### 2.3. Cell Number, Cell Viability, and Filopodia Outgrowth

Cell morphology after the various treatments was assessed using a bright-field Olympus BX43 microscope with the CellSens software (Version 3.2) (Olympus, Tokyo, Japan). Subsequently, cell number and viability were analyzed using a Cell Scepter electronic cell counter (EMD Millipore Corporation, Billerica, MA, USA) and the WST-1 assay (Roche Diagnostics, Taipei, Taiwan), respectively; moreover, the percentage of tBH-induced apoptotic cells was examined using 7-amino actinomycin D (7-AAD) (BD Pharmigen, Franklin Lakes, NJ, USA) staining and detected by a NucleoCounter ^®^ NC-3000™ fluorescence image cytometer (ChemoMetec, Alleroed, Denmark). The filopodia lengths of the treated cells were manually quantified using the ImageJ software (Version 1.48) (National Institutes of Health, Bethesda, MD, USA).

### 2.4. Quantitative Mitochondrial Morphology and Density

To visualize mitochondria, stained cells were mounted onto a perfusion chamber in a culture medium and imaged at 37 °C using an Olympus FluoView FV 1200 confocal microscope (Tokyo, Japan). The subtypes of mitochondrial morphology were quantified using the automatic morphological subtyping software (MicroP) developed by Peng et al. [22]. Briefly, the morphological classifications were made by analyzing the image features of the segmented mitochondrial objects, clustering them based on functional similarity, merging similar clusters, and defining distinct morphological subtypes based on the clustering results in the literature [22]. Finally, six morphological subtypes of mitochondria were identified, consisting of single and swollen globes, straight and twisting tubules, branched tubules, and loops. To calculate the proportion of globe mitochondria, the single globe and swollen globe shapes were summed and contrasted to the proportion of reticular mitochondria (i.e., the combined population of straight tubule, twisting tubule, and branched mitochondria). The analysis was performed on micrographs from three independent areas per group, and approximately 200–350 mitochondria from 6–8 cells in each image were analyzed semi-automatically. The mitochondrial counts were presented as the total number of mitochondria per cell according to data mentioned above.

### 2.5. Mitochondrial Function

Mitochondrial function was comprehensively assessed by measuring several parameters, including mitochondrial respiratory parameters: basal respiration, ATP-linked oxygen consumption (OCR), complex III-dependent OCR (CIII activity), mitochondrial DNA (mtDNA) copy number, and reactive oxygen species (ROS) generation (as a whole and in mitochondria (mtROS).

Mitochondrial respiration within saponin (1.25 ng/mL)-permeabilized cells were analyzed via high-resolution respirometry (Oxygraph-2k; Oroboros Instruments, Innsbruck, Austria) and expressed as the respiratory oxygen flow (pmol/second/million cells). The analysis started with the measurement of basal respiration (BR), which was defined as that recorded in cells in MiR05 buffer (Oroboros Instruments) in the absence of additional substrates or effectors; in contrast to ATP-linked production, which was defined as the oligomycin (1 μM)-mediated reduction of respiration in the presence of additional substrates [glutamate (G, 10 mM), malate (M, 2 mM), and ADP (2.5 mM)] (Sigma Aldrich, St Louis, MO, USA); and complex III activity, which was defined as the difference in the respiration change between the addition of duroquinone (DuroQ) (0.5 mM) (Sigma Aldrich) and antimycin A (AA, 5 μM) (Sigma Aldrich). Finally, the shutdown of the mitochondrial electron transport was executed by adding an inhibitor of the mitochondrial Complex IV, sodium azide (20 mM).

For the analysis of mtDNA copy number, DNA was extracted from cultured cells using a Qiagen DNeasy kit (Qiagen, Valencia, CA, USA). Next, quantitative PCR was performed using the SYBR Green PCR Master Mix (Roche Applied Science, Indianapolis, IN, USA) and an ABI Prism 7300 system (Applied Biosystems, Foster City, CA) with specific primer pairs to amplify the mtDNA-encoded nicotinamide adenine dinucleotide dehydrogenase subunit 1 (*ND1*) gene (forward primer: 5′–AACATACCCATGGCCAACCT–3′; and reverse primer: 5′–AGCGAAGGGTTGTAGTAGCCC–3′) and the nuclear DNA-encoded β-actin gene (as an internal control; forward primer: 5′–AGAAAATCTGGCACCACACC–3′; and reverse primer: 5′–CACCTTCTACAATGAGCTGCG–3′) from a total of 50 ng of DNA. The mtDNA copy number was determined based on the copy number ratio between these two genes (ND1/β-actin) [23].

ROS production was analyzed by measuring total ROS using the 2,7-dichlorofluorescein diacetate (DCFDA, Life Technologies) probe, whereas mitochondrial superoxide production was assessed using the MitoSox Red (Life Technologies) probe. The cells were incubated in a medium containing 10 μM DCFH-DA and 50 nM MitoSox Red at 37 °C for 15 min for staining in the dark. All stained cells were washed twice with PBS, resuspended in PBS, and kept on ice for immediate detection using a NucleoCounter ^®^ NC-3000™ fluorescence image cytometer.

### 2.6. Mitochondrial Motility

Mitochondrial transport imaging was carried out according to a previous study, with some modifications [24]. The live time-lapse imaging of cells treated with GFP-labeled mitochondria was performed using an Olympus FV1200 confocal laser scanning microscope at 37 °C with 5% CO_2_. The cells were imaged for 10 min with continuous recording. The acquired images were analyzed for mitochondrial motility using the CellSens (Olympus) software. Briefly, based on morphological criteria, for the analysis of the axonal initial segment, including the proximal and hillock regions, a 10–20-μm segment located at least 10 μm away from the soma was selected. The mitochondrial velocity duration time and the directionality of the movement, as obtained by measuring the angle of the slope, were analyzed in the kymographs generated by CellSens. The proportion of motile mitochondria was manually calculated by dividing the number of mitochondria moving faster than average by the total number of mitochondria based on the image sequences of the kymographs. At least 7–10 axons from 3–5 different cells were analyzed in each group. Considering that the axon extension in mutant cells was not obvious compared with that observed in WT cells, the mobility parameter was analyzed exclusively in the proximal region of axons.

### 2.7. Western Blotting and Protein Phosphorylation Antibody Array

Treated cells were rinsed with PBS and then suspended in RIPA buffer (Thermo Pierce, Rockford, IL, USA) supplemented with a complete protease and phosphatase inhibitor cocktail (EMD Millipore Corporation). The cells were kept on ice for 30 min and then homogenized. The extracts were centrifuged at 14,000× *g* for 20 min at 4 °C, and the supernatants were analyzed using the Bradford assay. Total proteins (20–30 μg) were separated by SDS–PAGE and transferred onto Immobilon-P membranes (EMD Millipore Corporation). The membranes were then incubated with the primary antibodies, as follows: anti-OPA1 (1:1000, BD Biosciences, Franklin Lakes, NJ, USA), anti-mitofusin 2 (Mfn2) (1:1000, NOVUS Biologicals, Littleton, CO, USA), anti-dynamin-related protein 1 (Drp-1) (1:500, EMD Millipore Corporation), anti-phospho-Drp-1-S616 (1:1000, Cell Signaling Technology, Danvers, MA, USA), anti-phospho-Drp-1-S637 (1:1000, Thermo Fisher Scientific, Inc., Waltham, MA, USA), anti-PTEN-induced kinase 1 (Pink1) (1:1000, NOVUS Biologicals), anti-Parkin (1:1000, Abcam, Cambridge, MA, USA), anti-phospho-p44/42 MAPK (ERK1/2) (Thr202/Tyr204) (1:1000, Cell Signaling Technology), anti-phospho-Akt (Ser473) (1:1000, Cell Signaling Technology), and anti-β-actin (1:1000, NOVUS Biologicals). The membranes were subsequently incubated with horseradish peroxidase (HRP)-conjugated secondary antibodies (Jackson ImmunoResearch, West Grove, PA, USA) at a dilution of 1:10,000. The signals were detected using an image-acquisition system (FUSION SL; Viber Lourmat, Marne-la-Vallee, France) and quantified using the Gel-Pro analyzer software (Version 3.0) (Media Cybernetics, Silver Spring, MD, USA).

The Human/Mouse Akt Pathway Phosphorylation Array C1 includes key proteins of interest in the relevant pathways (ERK1, ERK2, PRAS40, GSK3A, PTEN, GSK3B, RAF-1, mTOR, RPS6, Akt, p27, RSK1, AMPKa, p53, RSK2, BAD, P70S6K, 4E-BP1, and PDK1) and was performed according to the manufacturer’s instructions (RayBiotech, Norcross, GA, USA) [25]. Briefly, total protein preparation and HRP signal detection were carried out as described above. A total of 100 μg of the protein extracts from three independent cell lysates was added to each well and incubated for 24 h at 4 °C. Subsequently, the antibody array membranes were washed and incubated with the HRP-conjugated anti-rabbit IgG antibody included in the kit at room temperature for 2 h. The membranes were washed extensively before detection using chemiluminescence.

### 2.8. Protein Kinase A Activity

Protein kinase A (PKA) activity was examined using a PKA Colorimetric Activity Kit (Thermo Fisher Scientific, Waltham, MA, USA), including a PKA substrate-coated 96-well plate(s), a PKA standard, and an anti-phospho-PKA substrate antibody. Briefly, cell lysis was carried out in the presence of the supplied ATP, and phosphorylation was achieved by the immobilized PKA substrate. After a 90-min incubation followed by a wash, a rabbit antibody specific for the phospho-PKA substrate was bound to the modified immobilized substrate. Then, an antibody specific to rabbit IgG labeled with peroxidase was also added to the plate to bind to the rabbit anti-phospho-PKA substrate. After short incubation and wash steps, the substrate was added, and the absorbance was read at 450 nm using a CLARIOstar microplate reader (BMG LabTech, Ortenberg, Germany). The intensity of the color that developed was directly proportional to the amount of PKA in the samples and standards. All samples were read off the standard curve.

### 2.9. Statistics

The data are presented in the form of the mean ± standard deviation (SD) and were obtained from a minimum of three independent experiments. Associations between groups were assessed by Student’s *t*-test, and within-group differences were assessed by Bonferroni corrected post-hoc tests, which were performed using GraphPad Prism (GraphPad Software, La Jolla, CA, USA). Significance was set at *p* < 0. 05.

## 3. Results

### 3.1. Formoterol Increased the Viability of UQCRC1 Mutant Cells and Protected against Tertiary-Butyl Hydroperoxide-Induced Cell Damage in a β2-Adrenoceptor-Dependent Manner

The mutant cells exhibited hindered outgrowth (Figure 1a), diminished filopodia length (Figure 1b), and lower viability and proliferation compared to WT cells (Figure 1c). Following a 24-h formoterol treatment, there were significant improvements in these parameters, with a notable increase in cell activity and number (1.4-fold and 1.6-fold, respectively) compared to the disease DMSO group (Figure 1c). Filopodia outgrowth also significantly increased by up to 1.7-fold (Figure 1b). However, when formoterol was used in combination with propranolol, these positive effects were completely abolished (Figure 1a–c). No significant differences were observed in cell viability, growth, and filopodia outgrowth between the groups treated with formoterol plus propranolol and the DMSO group. These results indicate that formoterol’s response is mediated through the activation of β2AR.

Similar trends were observed in treated mutant cells subjected to tBH-induced oxidative damage (Figure 1d,e). Formoterol, compared to the DMSO group, significantly increased filopodia length by 1.9-fold under oxidative stress conditions through β2AR activation (Figure 1e). While this tBH dosage did not significantly induce cell apoptosis in WT cells, mutant cells showed a substantial 2.6-fold increase in the death rate compared to their non-treated counterparts (Figure 1f,g). Following a 24-h treatment, formoterol effectively reduced tBH-induced cell death by approximately 55% compared to the DMSO control (Figure 1g). Notably, the addition of propranolol nullified this effect, reinforcing the β2AR-dependent nature of formoterol’s impact on various cellular responses.

### 3.2. Formoterol Enhanced mtDNA Copy Number, Reduced ROS Levels, and Restored Mitochondrial Respiration and Complex III Activity, but Not ATP-Linked Respiration

To assess mitochondrial function comprehensively, various parameters were analyzed, including mtDNA copy number, ROS generation, and mitochondrial respiration in WT and mutant cells (Figure 2). Mutant cells exhibited distinct mitochondrial abnormalities compared to WT cells, including a significantly lower number of mtDNA copies (Figure 2a), elevated levels of total and mitochondrial reactive oxygen species (mtROS) generation (Figure 2b,c), and compromised cellular respiratory performance (Figure 2d, left panel). This was evident in markedly reduced basal respiration (BR), electron-transport-linked ATP respiration, and CIII-linked respiration, as depicted in Figure 2e. Formoterol consistently ameliorated these functional losses in mutant cells, enhancing mitochondrial substrate availability, except for ATP-linked respiration (Figure 2d,e). Propranolol addition nullified formoterol’s benefits except for mtROS production (Figure 2c), emphasizing the formoterol-mediated regulation of mitochondrial function via β2AR-dependent and independent actions.

### 3.3. Formoterol Improved the Transition from an Abnormal, Clustered Network to a More Tubular and Globular Mitochondrial Morphology

In contrast to WT cells, where mitochondria were evenly distributed in the cytosol, mutant cells showed abnormalities in both mitochondrial organization and morphology (Figure 3a, left panel). Mutant cells exhibited an excessive clustering of mitochondria in mutant cells, resulting in the formation of large mitochondrial networks around nuclei. It was evidently illustrated in the right panel of Figure 3a, marked by a prevalence of reticular mitochondria exhibiting branching (purple tinge), twisting (orange tinge), and loop structures (red tinge), contrasted with a reduced number of tubular mitochondria (green tinge) and globe mitochondria, including single structures (blue tinge) and swollen structures (yellow tinge). Formoterol effectively rectified this imbalance in mitochondrial morphology, inducing a more even distribution of mitochondria within the cells in mutant cells, resembling the pattern observed in WT cells (Figure 3a). The quantification of clustering mitochondrial morphologies into the reticular network, tubule, and globe subtypes (Figure 3b) revealed that formoterol reduced the fraction of the reticular mitochondria subtype by approximately 50% compared to the DMSO group while increasing the fractions of the tubular and globe mitochondria subtypes. The addition of propranolol negated this effect, indicating that formoterol’s actions on mitochondrial morphology were specifically mediated by β2AR activation (Figure 3b).

### 3.4. Formoterol Regulated Mitochondrial Fusion–Fission Balance through β2-Adrenoreceptor Activation and Was Linked to Counter-Regulation of ERK and Akt Signals

Mutant cells, compared to WT cells, showed a heightened preference for mitochondrial fusion over division, as depicted in Figure 4. The results from Western blot analysis (Figure 4a) and the quantification of proteins associated with mitochondrial dynamics (Figure 4b) showed a substantial elevation in Mfn2 levels (excluding OPA1) and a notable decrease in both Drp-1 expression and its phosphorylation at S616 (Drp-1 S616), a factor recognized for promoting mitochondrial fission. It was accompanied by a distinct dephosphorylation of Drp-1 at S637, a factor recognized for inhibiting mitochondrial fission. Formoterol treatment significantly reversed the aforementioned alterations in Mfn2, Drp-1, and Drp-1 S616 phosphorylation in mutant cells (Figure 4a), showing a significant difference compared to the DMSO control group (Figure 4b). Remarkably, dephosphorylation of Drp-1 at S637 was also distinctly enhanced by formoterol treatment. Furthermore, mutant cells exhibited inhibition of full-length Pink1 (L-form) and no impact on the Parkin protein compared to WT cells (Figure 4a). Formoterol treatment resulted in a significant upregulation of both the full-length Pink1 and the cleaved Pink1 (S-form) proteins, alongside a simultaneous downregulation of the mitophagy-related protein Parkin (Figure 4b). It revealed that the formoterol-induced mitochondrial fission did not result in mitochondrial degradation in mutant cells. Overall, the comprehensive regulation of mitochondrial dynamic-related proteins by formoterol was shown to be mediated through β2AR activation. This was evident from the reversal of these performance actions upon the addition of propranolol, except for Parkin protein, where no significant differences were observed between the formoterol treatments with and without propranolol.

To elucidate the downstream pathway of β2AR activation linking the regulation of mitochondrial dynamic, an intercellular signaling array with MAPK phosphorylation antibodies was employed (Figure 5a). Mutant cells displayed elevated levels of Akt phosphorylation at S473, in contrast with the phosphorylation status of ERK1/2 at T202/Y204 and Y185/Y187, compared with WT cells (Figure 5a, left panel). Formoterol effectively counter-regulated the expression of Akt and ERK phosphorylation, and this specific regulation was demonstrated to be dependent on β2AR activation, as demonstrated by the abolished regulation upon cotreatment with propranolol (Figure 5a, left panel). Furthermore, a comparison of pathway-related phosphorylated proteins integrated into the array was conducted for each set of groups. The differences in the expression of the corresponding proteins were listed individually in the right panel of the bar graph depicted in Figure 5a (for the raw data, see Appendix A). Among the Raf–ERK signaling molecules involved in the positive regulation of formoterol, the induction of 4E-BP1 was even higher than that of ERK1/2. In contrast, formoterol negatively regulated the Akt–RSK–S6-kinase signaling-related molecules (Figure 5a, right panel). Consistent findings of the restoration of ERK signaling and the inhibition of Akt activation in mutant cells by formoterol were confirmed through Western blot analysis (Figure 5b). Furthermore, Furthermore, formoterol treatment showed no effect on the elevated PKA activity in mutant cells when compared to WT cells (Figure 5c).

Figure 5d depicts the pathway diagram corresponding to the regulatory relevance of mitochondrial dynamics through the formoterol-induced activation of β2AR. The improvement in the dynamic balance afforded by formoterol was attributed to its simultaneous promotion of the ERK pathway and inhibition of the Akt pathway. This led to an increase in the ratio of Drp-1 S616/Drp-1 S637 in mutant cells, resulting in the favoring of mitochondrial fission over fusion. Simultaneously, the inhibition of Akt overactivation may contribute to the downregulation of Mfn2 expression, therefore further enhancing the manifestation of the aforementioned effects.

### 3.5. Formoterol Increased the Efficiency of Mitochondrial Anterograde Transportation and Its Mobility

Time-lapse recordings of MitoGFP-labeled mitochondria, together with corresponding kymographs, were utilized to analyze mitochondrial transport in proximal axons between adjacent cells (Figure 6). The representative kymographs provided in Figure 6a illustrated the bidirectional movement of signals as diagonal lines, revealing more active and abrupt movements of mitochondria in WT cells (Appendix A) compared to mutant cells (Appendix A). In addition, WT cells also had a higher velocity for anterograde transport (mean velocity: WT cells, 51.2 ± 0.013 nm/s; vs. UQCRC1 mutant cells, 7.2 ± 0.002 nm/s; Figure 6b). The rate of mitochondrial retrograde transport was similar in both WT and mutant cells (Figure 6b). Moreover, a substantial decrease in the fraction of motile mitochondria was noted, showing a 52.6% reduction in anterograde transport and a 48.6% reduction in retrograde transport in mutant cells (Figure 6c). Formoterol treatment significantly enhanced mitochondrial mobility (Appendix A), elevating anterograde movement velocity by 2.1-fold (mean velocity, 25.6 ± 0.009 nm/s) and increasing the motile ratio by 1.5-fold compared to the DMSO group (Figure 6c). However, formoterol did not exert significant effects on the frequency and rate of mitochondrial retrograde movement (Figure 6b,c). The induction of mitochondrial velocity and mobility by formoterol was consistently nullified by the addition of propranolol (Figure 6a–c) (Appendix A). Additionally, individual frames from the time-lapse images presented in Figure 6d further illustrated that mitochondria in the WT and formoterol-treated groups displayed significant forward displacement in axons during the time-lapse sequence (the anterograde direction is indicated by the arrow in Figure 6d). In contrast, stationary mitochondria were observed in the mutant cells and those treated with formoterol plus propranolol or propranolol alone (Figure 6d). These findings indicated that the enhancement in mitochondrial velocity and mobility induced by formoterol occurred through β2-adrenoceptor activation.

## 4. Discussion

The effect of formoterol on mitochondrial function has been validated in various diseases, including traumatic brain injury [26], spinal cord injury [27], diabetic kidney disease [28], and acute kidney injury, with those studies providing compelling evidence in support of the use of β2AR ligands for therapeutic mitochondrial biogenesis. This study further demonstrated the therapeutic efficacy of formoterol in PD and explored its impact on mitochondrial dynamics. Abnormalities in fusion and fission processes, crucial in PD pathogenesis [14], are addressed. Formoterol treatment enhanced cell viability, offering protection against ROS-induced cell death. Beyond restoring mitochondrial respiration, it rebalanced mitochondrial dynamics, normalizing the network by segregating healthy mitochondria and facilitating anterograde movement in UQCRC1-mutant cells. Efficacy depended on B2AR activation involving downstream cascades: ERK activation and Akt signaling inhibition. This study underscores formoterol’s pivotal role as a mitochondrial dynamic balance regulator, positioning it as a promising therapeutic candidate for PD. On the other hand, understanding the role of mitochondrial homeostasis provides critical insights into the debates surrounding β2AR agonists in PD therapy. By addressing mitochondrial dysregulation, β2AR agonists may offer neuroprotective effects that go beyond symptomatic relief. Clinical trials evaluating β2AR agonists should consider assessing mitochondrial function as a primary outcome measure to elucidate their therapeutic efficacy fully. Moreover, stratifying PD patients based on biomarkers of mitochondrial dysfunction and dynamics dysfunction mitochondrial dysfunction and biomarkers, may help identify subpopulations most likely to benefit from β2AR agonist therapy, therefore resolving uncertainties regarding patient selection and treatment response.

Our study shows that the inhibition of PKA activity by propranolol cotreatment abolished the beneficial effects of formoterol on functional recovery and oxidative stress defense in mutant cells. We propose that the initial PKA activation observed in UQCRC1-mutant cells may serve as a player in neuroprotective mechanisms, as supported by previous research [29,30]. However, restoration of the mitochondrial defects caused by UQCRC1 mutation does not affect PKA activity. It indicates the existence of a mitochondrial function-independent regulation in PKA regulation. For example, calcium ions can activate calmodulin, which subsequently activates adenylate cyclase, leading to elevated cyclic adenosine monophosphate (cAMP) levels and PKA activation. Furthermore, a high level of PKA activity phosphorylates enzymes involved in metabolic pathways and changes metabolic fluxes, which may indirectly affect ATP levels by altering the availability of substrates and intermediates involved in ATP synthesis. Hence, it is plausible that the formoterol treatment had no impact on ATP-linked respiration in mutant cells. Incidentally, although regulating mitochondrial dynamics typically requires significant energy consumption, this is not the case with formoterol’s rescue of mitochondrial dynamic abnormalities stemming from mutations in UQCRC1, which impair CIII function in cells. As mitochondrial fragmentation triggered by CIII activity loss operates through energy-independent regulation, it stands in contrast to CV [31]. In particular, our prior research revealed that the mutation in UQCRC1 had minimal effect on the function of CI, CII, or CIV, aside from CIII [17]. This also emphasizes that the relationship between mitochondrial dynamics and mitochondrial oxidative phosphorylation can be dissociated in certain scenarios [32].

PKA activity is primarily regulated by cAMP levels. Elevated cAMP not only boosts PKA activity but also activates adenosine monophosphate-activated protein kinase (AMPK). PKA signaling has been confirmed to induce neuroprotective mitochondrial restructuring into an interconnected network [29]. Otherwise, both PKA and Akt can phosphorylate AMPK to inhibit its activity, resulting in the suppression of mitochondrial fission through Drp-1 phosphorylation [33] and mitochondrial transport in neural axons [34]. Based on our findings, which revealed that formoterol restored mitochondrial fission and anterograde transport by inhibiting Akt phosphorylation while leaving PKA activity unaffected, we propose that the formoterol-mediated inhibition of Akt activation, rather than PKA suppression, plays a pivotal role in regulating the aforementioned mitochondrial dynamics. Further clarification is needed to understand the impact of formoterol on the regulation of AMPK related to mitochondrial dynamics. On the other hand, the consistent finding of extensively interconnected mitochondrial networks observed in UQCRC1-mutant cells is also found in UQCRC2-mutant fibroblasts derived from patients with severe encephalomyopathy [35]. It is well known that UQCRC1 and UQCRC2 are both subunits of CIII that play essential roles in the structure and function of CIII [35]. Unlike UQCRC2-mutant cells, which display an expanded reticular network of mitochondria, more fragmented mitochondria are present [35]. We discovered that UQCRC1 mutation significantly disrupted mitochondrial dynamics, shifting towards hyperfusion, as shown by a notable increase in the reticular network of mitochondria and a decrease in fragmented mitochondria. Reducing mitochondrial stress with formoterol in mutant cells indeed rebalances mitochondrial dynamics and prevents excessive reticular network formation. Indeed, the persistent excessive fusion of mitochondria disrupts both their distribution and turnover, leading to locomotor defects in Drosophila models of Charcot–Marie–Tooth disease type 2A neuropathy [36]. Furthermore, prolonged elongation of mitochondria hinders mitophagy, the process responsible for clearing dysfunctional mitochondria, which in turn diminishes overall mitochondrial function [37,38]. It also disrupts the distribution and transport of mitochondria within cells [32,39], as demonstrated in the present study, and interferes with their interactions with other cellular structures, potentially impacting cellular signaling and response [32].

Formoterol treatment increased Drp-1 activity via phosphorylated Drp1 S616 and dephosphorylated Drp-1 S637 to provide a fine-tuned control of mitochondrial fission and restore an imbalanced mitochondrial dynamic. Drp1, a key regulator of mitochondrial fission, undergoes site-specific phosphorylation. Phosphorylation at Drp1 S616 enhances mitochondrial fission, while phosphorylation at Drp1 Ser637 inhibits this process, triggered by various signaling pathways [40,41]. A current study indicates that the Akt-1 pathway coordinates the concurrent dephosphorylation of Drp1 S616 and phosphorylation of Drp1 S637, leading to the inhibition of mitochondrial fission. Conversely, the promotion of mitochondrial fission through the counter-regulation of Drp1 phosphorylation is mediated by the MEK1-ERK pathway [40]. The two axes of the Akt1–Drp1 and MEK1–ERK-Drp1 pathways can be switched to remodel the mitochondrial dynamics in somatic cell reprogramming [40]. Our finding corroborates this finding: mutant cells with impaired mitochondrial fission exhibit reduced ERK signaling and increased Akt activity compared to WT cells. Formoterol treatment restores mitochondrial dynamics towards fission by concurrently enhancing ERK signaling for Drp-1 S616 phosphorylation and inhibiting Akt signaling for Drp-1 S637 phosphorylation. Furthermore, mutant cells show elevated levels of Mfn2 proteins, in contrast to Drp-1. The physiological levels of Mfn2 expression are strongly correlated with the Akt signaling pathway, to promote mitochondrial fusion [42]. Hence, imbalance in mitochondrial dynamics, favoring fusion over fission, is predominantly attributed to the overactivation of the Akt pathway rather than ERK signals in mutant cells.

Several of the known genes associated with the familial forms of PD are involved in the regulation of mitochondrial function, including Parkin and Pink1 [43]. The downregulation of Pink1 affects the mitochondrial fusion–fission machinery and sensitizes mitochondria to neurotoxins in dopaminergic cells [44]. Pink1 not only phosphorylates Drp-1 S616 to activate mitochondrial fission [45] but also phosphorylates Mfn2 to promote Parkin recruitment for mitophagy, which is a selective process that removes damaged mitochondria [46]. Loss of Pink1 impairs mitochondrial function, leading to mitochondrial dysfunction, increases oxidative stress, and compromises cellular energy production [47]. This is in line with our observation of reduced Pink1 expression in mutant cells. Interestingly, in cells treated with formoterol, both full-length Pink1 and its 52-kDa cleaved form [48] were upregulated, whereas the levels of Parkin and Mfn2 were significantly decreased. We suggest that the formoterol-induced upregulation of Pink1, with consequent mitochondrial fission, does not activate mitophagy. Since Pink1 also regulates mitochondrial fission by phosphorylating Drp1 S616 through a mechanism independent of mitophagy [45]. Moreover, when Pink1 is cleaved at the inner mitochondrial membrane, it retro-translocates to the cytosol, inhibiting Parkin translocation to mitochondria and suppressing mitophagy [49]. Although the precise role of cleaved Pink1, induced by formoterol, in neuronal functions, remains unclear, in healthy mitochondria, it plays vital extramitochondrial roles crucial for neuronal development, survival, synaptogenesis, and plasticity, with significant implications for PD [50,51]. Thus, we suggest that the restoration of Pink1 could play a multifaceted role in supporting neuronal function in the context of the benefits of formoterol.

Formoterol has been shown to decrease the glucose-induced imbalance in mitochondrial dynamics and restore mitochondrial homeostasis in the context of renal proximal tubule cells under diabetic conditions [28]. A similar regulatory role of formoterol was found in the treatment of UQCRC1-mutant cells, resulting in PD, although the specific mechanisms underlying the regulation of mitochondrial homeostasis appear to be dependent on various cellular contexts. Moreover, our findings provide further evidence of the beneficial effects of formoterol, as it significantly promoted anterograde mitochondrial transport and increased mitochondrial mobility. Several studies have explored strategies aimed at enhancing anterograde mitochondrial transport as a means of neuroprotection to manage mitochondrial dysfunction-related neurodegenerative disorders [52,53,54]. The promotion of mitochondrial anterograde movement facilitates the transport of mitochondria toward axon terminals to support the energy demands required for synaptic transmission and neuronal signaling [53,54], maintain mitochondrial health via the overall quality control of mitochondria within neurons [54], and ensure proper neuronal functions, such as calcium regulation, excitability, and neurotransmission [52,53]. Additionally, the effects of formoterol on healthy SH-SY5Y cells have also been recognized as a crucial factor in neuroprotection and neuroregeneration [55]. This was achieved through the increase in brain-derived neurotrophic factor (BDNF), which is well known for enhancing neuronal survival, differentiation, and synaptic plasticity, along with its receptor TrkB [55]. However, further investigation is necessary to completely understand its mechanisms in neural regulation.

While preclinical evidence indicates promise for formoterol’s neuroprotective potential in PD, clinical data regarding the disease-modifying effects of β2AR agonists remains inconclusive. This uncertainty stems from various challenges, including the need to optimize dosing regimens, comprehend the long-term safety profile of formoterol in PD patients, and identify dependable biomarkers for monitoring treatment response and disease progression, which remain unresolved. Further research is essential to uncover the mechanistic basis of formoterol’s neuroprotective effects and determine its efficacy as a disease-modifying therapy for PD using human-relevant systems. Patient-derived cell models, such as induced pluripotent stem cells from PD, offer a unique platform for further assessing formoterol’s efficacy and safety, closely mirroring the pathophysiological mechanisms of abnormal mitochondrial homeostasis in PD.

## 5. Conclusions

In summary, by activating B2AR, formoterol not only enhanced mitochondrial function but also corrected the imbalance in mitochondrial dynamics and transportation. This correction contributed to improving the disruption of mitochondrial homeostasis in familial PD caused by mitochondrial genetic defects. The regulatory mechanism involves the restoration of ERK signaling and the inhibition of Akt overactivity, contingent on the activation of β2AR. Formoterol’s neuroprotective effects, attributed to its regulation of mitochondrial dynamic balance to uphold mitochondrial homeostasis, suggest promising potential for treating other neurodegenerative diseases with a similar etiology. Moving forward, it is essential to further validate the specificity of formoterol’s impact through β2AR on mitochondrial dynamics by integrating experiments involving β2AR knockdown or overexpression, alongside confirming this regulatory mechanism in vivo.

## Figures and Tables

**Figure 1 biology-13-00231-f001:**
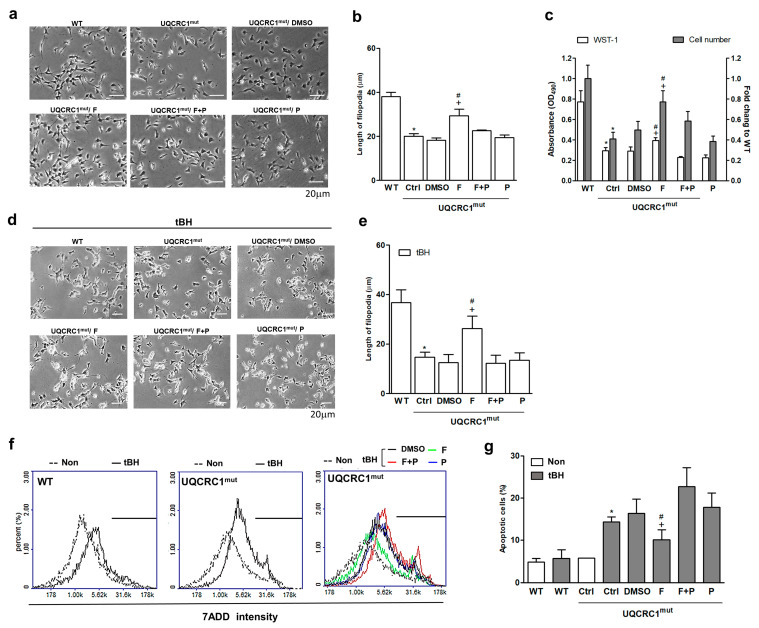
Cell viability and filopodia outgrowth in the absence or presence of tertiary-butyl hydroperoxide (tBH) stimulation. Cell morphology was observed after treatments with or without tBH stimulation for 24 h (10 μM) (**a**,**d**). The filopodia length was quantified by image processing using ImageJ (**b**,**e**). Cell viability and cell proliferation were determined by WST-1 analysis and counted with a Coulter counter (**c**). The tBH-induced cell death was measured using 7-AAD staining and flow cytometry analysis (**f**), then quantified (**g**). * *p* < 0.05 vs. the WT group or the treated Ctrl group. + *p* < 0.05 vs. the DMSO group. # *p* < 0.05 vs. the F + P group. Data are presented as the mean ± SD. WT: wild-type; Ctrl: control; DMSO, dimethylsulfoxide; F, formoterol; F + P: formoterol plus propranolol; P: propranolol; UQCRC1mut: mutation of the ubiquinol-cytochrome c reductase core protein 1 gene; 7AAD; 7-aminoactinomycin D; tBH: tertiary-butyl hydroperoxide. *N* = 6.

**Figure 2 biology-13-00231-f002:**
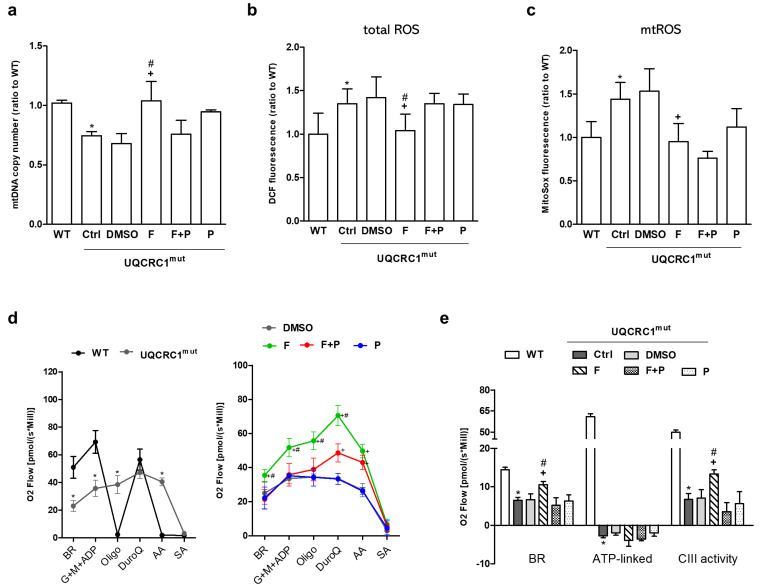
Mitochondrial function and generation of reactive oxygen species. Mitochondrial function was comprehensively assessed after 24h treatments by analyzing the mtDNA content (DNA copy number) (**a**), the level of reactive oxygen species in total (total ROS) (**b**) and in mitochondria (mtROS) (**c**), and mitochondrial respiration (**d**,**e**). Mitochondrial oxygen consumption was measured in permeabilized cells using different substrates and inhibitors of the respiratory chain complexes; the comparison between WT and mutant cells (left panel) or between mutant cells with different treatments (right panel) is shown (**d**). The induced oxygen consumption (oxygen flux) was integrated and quantified according to different approaches to provide an indirect measurement of mitochondrial activity, including basal respiration, ATP−linked respiration (oligomycin-mediated reduction), and CIII−linked respiration (respiratory difference between duroquinone, a CIII substrate, and antimycin A, a CIII inhibitor) (**e**). * *p* < 0.05 vs. the WT group. + *p* < 0.05 vs. the DMSO group. # *p* < 0.05 vs. the F + P group. Data are presented as the mean ± SD. WT: wild-type; Ctrl: control; DMSO, dimethylsulfoxide; F, formoterol; F + P: formoterol plus propranolol; P: propranolol; UQCRC1mut: mutation of the ubiquinol-cytochrome c reductase core protein 1 gene; BR: basal respiration, G: glutamate, M: malate, Oligo: oligomycin; DuroQ: duroquinone, AA: antimycin A, SA: sodium azide. *N* = 3.

**Figure 3 biology-13-00231-f003:**
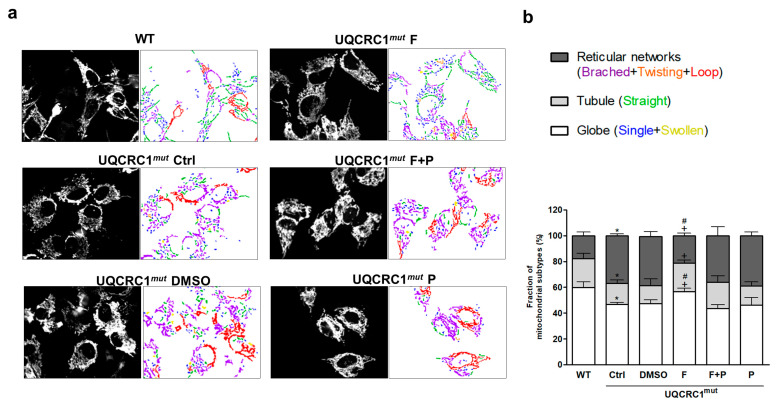
Mitochondrial morphology and distribution. The morphology of GFP-labeled mitochondria was observed after 24-h treatments ((**a**), left panel). The classification of mitochondrial morphology was further analyzed using automatic MicroP software and was presented by the labeling of different color-coded subtypes of mitochondria to show the morphological composition of each group of representative cells ((**a**), right panel). The six distinct mitochondrial subtypes were labeled with different colors ((**a**), right panel) and were classified as class three, as follows: globe, including single (blue) and swollen (yellow); tubule, including straight tubule (green) and reticular networks [including branched tubule (purple), twisting tubule (orange), and loop (red)] (**b**). The percentage of each mitochondrial subtype among the total mitochondrial population was calculated and integrated into the three categories, as described above (**b**). * *p* < 0.05 vs. the WT group. + *p* < 0.05 vs. the DMSO group. # *p* < 0.05 vs. the F + P group. Data are presented as the mean ± SD. WT: wild-type; Ctrl: control; DMSO, dimethylsulfoxide; F, formoterol; F + P: formoterol plus propranolol; P: propranolol; UQCRC1mut: mutation of the ubiquinol-cytochrome c reductase core protein 1 gene.

**Figure 4 biology-13-00231-f004:**
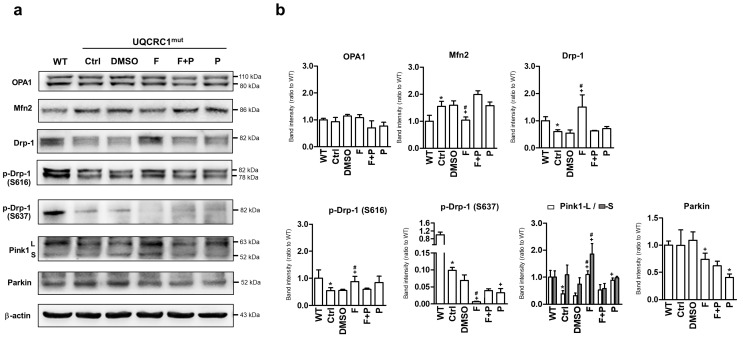
Expression of mitochondrial dynamic-related proteins. The level of mitochondrial fusion-relate (OPA1 and Mfn2), fission-related [total and phosphorylated form of Drp-1 at serine 616 (S616) and 637 (S637)], and mitophagy-related [Long (L) and short form (S) of Pink1 and Parkin)] proteins were analyzed (**a**) by Western blotting and quantified in both WT cells and mutant cells with or without 24-h treatments. Target proteins were quantified by normalizing to β-actin and were presented as the fold change relative to the WT group (**b**). * *p* < 0.05 vs. the WT group. + *p* < 0.05 vs. the DMSO group. # *p* < 0.05 vs. the F + P group. Data are presented as the mean ± SD. WT: wild-type; Ctrl: control; DMSO, dimethylsulfoxide; F, formoterol; F + P: formoterol plus propranolol; P: propranolol; UQCRC1mut: mutation of ubiquinol-cytochrome c reductase core protein 1 gene. N = 3.

**Figure 5 biology-13-00231-f005:**
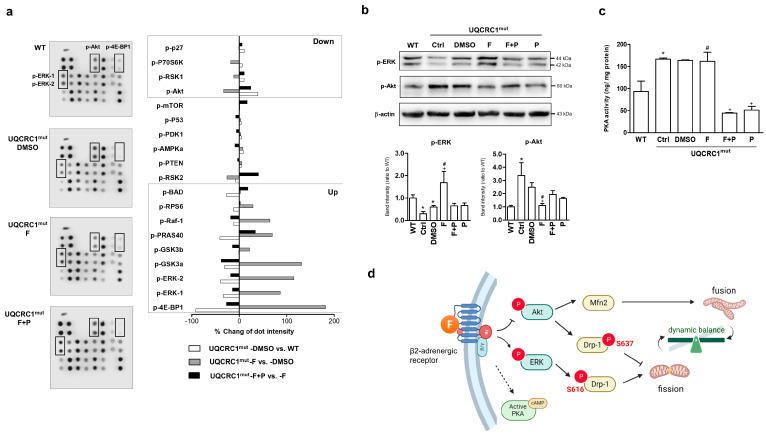
Expression of the downstream related signaling in activation of the β2 adrenergic receptor. T-related regulatory signaling of ERK and Akt phosphorylation was analyzed using a phospho-kinase array of the pools of cell lysates from three independent experiments (**a**). Left panel: immunoblotted membranes (highlighted squares: spots with obvious differences in phosphorylation levels); left panel: quantifications of phosphorylation levels of all spots. The expression differences between different groups were compared (UQCRC1mut−DMSO vs. WT; UQCRC1mut−F vs. UQCRC1mut−DMSO; UQCRC1mut−F + P vs. UQCRC1mut−F), and the reverse regulation direction (up- or downregulation) between inter-groups, UQCRC1mut−DMSO vs. WT; UQCRC1mut−F vs. UQCRC1mut−DMSO, was indicated in the squares (A, right panel). The ERK and Akt phosphorylation was further confirmed by Western blotting, quantified by normalizing to β-actin, and were presented as the fold change relative to the WT group (**b**). The activity of classic protein kinase A (PKA) was measured using a colorimetric activity assay (**c**). A path diagram was constructed based on the above findings to illustrate the regulatory relationship between the mitochondrial dynamic machinery and the activation of the β2 adrenergic receptor by formoterol (**d**). Formoterol treatment rebalanced mitochondrial dynamics by promoting fission and inhibiting abnormal hyperfusion in mutant cells. The regulatory machinery was associated with activating downstream ERK signaling and inhibiting Akt signaling (solid line) rather than being regulated by PKA (dotted line). * *p* < 0.05 vs. the WT group. + *p* < 0.05 vs. the DMSO group. # *p* < 0.05 vs. the F + P group. Data are presented as the mean ± SD. WT: wild-type; Ctrl: control; DMSO, dimethylsulfoxide; F, formoterol; F + P: formoterol plus propranolol; P: propranolol; UQCRC1mut: mutation of ubiquinol-cytochrome c reductase core protein 1 gene; cAMP, cyclic adenosine monophosphate). *N* = 3.

**Figure 6 biology-13-00231-f006:**
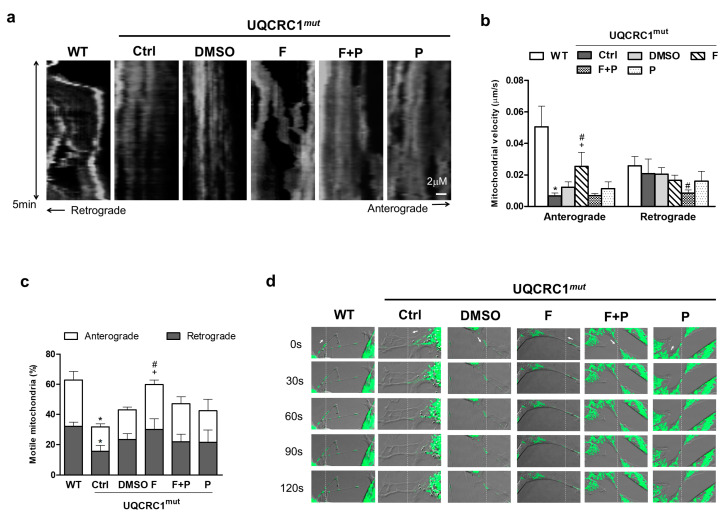
Mitochondrial movement and mobility. Representative kymographs showing the movement of the mitochondria-labeling GFP signal in WT cells or mutant cells with or without 24-h treatments. Vertical lines were observed for stationary mitochondria, whereas moving mitochondria displayed diagonal lines, indicating their motion in either the anterograde or retrograde direction (**a**). Mitochondrial transport was quantified by measuring mitochondrial velocity (**b**) and calculating the proportion of motile mitochondria exceeding the average velocity in relation to the total mitochondrial population (**c**). Representative time-lapse images specifically showed the mitochondria undergoing anterograde movement (direction indicated by arrows) from their initial position (dashed line, (**d**)). * *p* < 0.05 vs. the WT group. + *p* < 0.05 vs. the DMSO group. # *p* < 0.05 vs. the F + P group. Data are presented as the mean ± SD. WT: wild-type; Ctrl: control; DMSO, dimethylsulfoxide; F, formoterol; F + P: formoterol plus propranolol; P: propranolol; UQCRC1mut: mutation of ubiquinol-cytochrome c reductase core protein 1 gene.

## Data Availability

The datasets generated during and/or analyzed during the current study are not publicly available as consent for publication of raw data was not obtained from study participants, but they are available from the corresponding author upon reasonable request.

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
