# Peer review of "Formoterol Acting via β2-Adrenoreceptor Restores Mitochondrial Dysfunction Caused by Parkinson’s Disease-Related UQCRC1 Mutation and Improves Mitochondrial Homeostasis Including Dynamic and Transport"

_biology, 2024, doi:10.3390/biology13040231_

Round 1

Reviewer 1 Report

Comments and Suggestions for Authors

please see file

Comments on the Quality of English Language

Minor editing of English language required

Reviewer 2 Report

Comments and Suggestions for Authors

The manuscript by Chang and colleagues is interesting and presents a comprehensive investigation of Formoterol, a β2-adrenergic receptor agonist, using SHY5Y cell culture cells and the UQCRC1 mutant linked to familial parkinsonism, Overall, Formoterol shows promise as a therapeutic candidate for Parkinson's disease by acting as a regulator of mitochondrial dynamics balance.

In the graphical abstract, the authors featured a neuron cell; however, they did not mention the usage of differentiated SHY5Y cells. Undifferentiated SHY5Y cells are not considered neuron-like cells. Furthermore, in the method section (2.3): "Cell number, cell viability, and neurite outgrowth," the use of terms like "neurites" or "neurons" should be avoided as it is misleading to the reader. Neurons and neurites are well-defined biological entities that are not represented by these cells. Please change the wording accordingly throughout the manuscript. Outgrowths or filopodia could be used instead.

Major: In Figure 1, in panels b, c, e, and f/g, it is important to add the same experimental tests for the WT group to match those of the mutant group. Those are crucial controls that are missing, i.e., please provide matching data sets of WT cells that have been subjected to the same treatments (additionally, including an antioxidant like NAC as a negative control for tBH would enhance the experimental design – something the authors could consider including). The authors are advised to rephrase references to "neurites" as "filopodia" in order to prevent misinterpretation of the results.

In Figure 2, attention should be directed towards Figure 2c, where the addition of Formoterol + Propranolol seemingly reduces mtROS without achieving significance compared to WT. It seems Propranolol did not neutralize the effect of Formoterol in reducing mtROS. The total ROS change is then non-mitochondrial related? How is this explained? Furthermore, in Figure 2d, the lack of change observed when oligomycin is added to UQCRC1mut suggests that mitochondrial function is not rescued, indicating that mitochondria are still uncoupled after Formoterol treatment.

Changes in terminology are suggested, such as replacing "ATP-linked production" with "ATP-linked respiration" (line 317) and "mitochondrial respiration" with "mitochondrial oxygen consumption" (line 312). Also, change "CIII-activity" to "CIII-linked respiration." Line 307, "except for mtROS expression," do the authors mean mtROS change? Please adjust the terminology throughout the manuscript.

For Figures 2d and 2e, significance measurements should be added.

The graphs start at zero in Fig. 2d. It would be best to not show a line between the x- and y-axis interception and the first data points since the experiment starts with the basal respiration measurements.

Major: In Fig. 2d and 2e Respiration is recorded in negative values (for 3 values in 2d and 1 value in ) – how is this possible? There must be a major problem here.

After addition of oligomycin the WT cell show negative oxygen consumption rates – this should go down quite a bit but not into the negative range. In addition, the mutant cells do not respond at all to oligomycin treatment – how is this possible? This entire Figure 2d raises serious experimental concerns.

Furthermore, the inconsistency between the change in CIII-linked respiration between F+P and P in Figure 2d and Figure 2e should be clarified, or the experiments should be repeated: In Figure 2e, the change of CIII-linked respiration of F+P is higher than P, but in Figure 2e, F+P is smaller than P for CIII-linked respiration.

The authors are encouraged to refine the statistical reporting, using different symbols for each comparison grouping. Most people consider ** to indicate P < 0.01 (in the manuscript, * and ** are used for different groups and not different significant level indication). Please use different symbols for each comparison and apply this change to all figure’s quantifications in the manuscript.

Figure 3: Please provide an explanation for morphological classifications used.

Figure 5c: It is suggested to explore the potential impact of WT+F+P on WT's PKA activity.

Figure 5d. If possible please provide a more detailed figure for the proposed pathway.

Reviewer 3 Report

Comments and Suggestions for Authors

The manuscript titled 'Formoterol acting via β2-adrenoreceptor restored mitochondrial dysfunction caused by Parkinson's disease-related UQCRC1 mutation and improves mitochondrial homeostasis including dynamic and transport' by Chang et al. explores the therapeutic potential of formoterol, a β2-adrenergic receptor agonist, in mitigating mitochondrial dysfunction associated with Parkinson's disease through the modulation of mitochondrial dynamics. The  focus of the study on the specific UQCRC1 mutation and its detailed investigation into the cellular mechanisms underlying the action of formoterol, represents a significant contribution to the field of neurodegenerative disease research. The manuscript is well-structured and presents a comprehensive analysis of the effects of formoterol on mitochondrial homeostasis, including dynamics and transport in a PD-related cellular model. While the study advances our understanding of potential benefits of formoterol in PD treatment, there are several areas where further clarification and additional data could enhance the impact and clarity of the findings. Below, I outline my major and minor comments, which I believe will help strengthen the manuscript and better position it within the current body of literature.

Major Comments:

Introduction: The hypothesis concerning  impact of formoterol on mitochondrial dynamics within PD models requires further expansion. It would be beneficial to more explicitly connect β2AR activation by formoterol with anticipated outcomes on mitochondrial function and the viability of neuronal cells. This clarification will help readers understand the foundational premise of your research.

Results and Discussion: Additional evidence or discussion on the specificity of the action of formoterol via β2AR in affecting mitochondrial dynamics is needed. Consider incorporating experiments involving β2AR knockdown or overexpression to solidify your claims about the specificity of formoterol's mechanism of action (if possible).

Discussion: The manuscript would greatly benefit from a deeper exploration of the molecular mechanisms by which β2AR activation influences mitochondrial dynamics. Specifically, elucidate how ERK and Akt signaling pathways are involved in modulating mitochondrial fission/fusion balance and providing neuroprotection in PD models.

Discussion: The potential long-term effects of formoterol treatment, along with its safety profile, should be addressed. This is particularly important considering formoterol's primary use in treating asthma and COPD. The implications of systemic β2AR activation on PD patients warrant further discussion.

Minor Comments:

Enhance the clarity of the introduction by concisely summarizing the current debates surrounding the role of β2AR agonists in PD and how this study contributes to resolving those debates.

Methods: Provide a clear rationale for selecting 1 µM formoterol and 100 µM propranolol concentrations, based on literature review or preliminary data.

Experimental Limitations: Discuss any limitations of your experimental approach and suggest future research directions, such as in vivo studies or investigations into the effects of formoterol on human-derived neuronal cultures.

Conclusion: Mention potential future research directions, including the exploration of formoterol’s effects in vivo or in patient-derived cell models, to further validate and extend the current findings.

Comments on the Quality of English Language

The quality of English language throughout the manuscript is commendable, allowing for smooth reading and comprehension of the study's objectives, methods, results, and conclusions. However, minor editorial revisions could help eliminate occasional grammatical errors and improve sentence structure in a few sections to ensure consistency and enhance the overall readability of the document.

Round 2

Reviewer 2 Report

Comments and Suggestions for Authors

To further enhance the quality of the manuscript, the authors should consider adding discussion and citations regarding the effect of Formoterol on WT cells, as conducting new experiments was not possible for them.

Author Response

Dear reviewer,

I appreciate your efforts to enhance the quality of the paper. Following your advice, we have included the relevant discussion in the revised manuscript, which is indicated in blue color in the text of the revision. Please refer to the explanation provided in the response below. Thank you.

Point-by-point response to Comments and Suggestions for Authors

Reviewer comments:

Reviewer 2

To further enhance the quality of the manuscript, the authors should consider adding discussion and citations regarding the effect of Formoterol on WT cells, as conducting new experiments was not possible for them.

Reply: According to your suggestion, the following discussion content has been added to the modified manuscript (lines 616-622): Additionally, the effects of Formoterol on healthy SH-SY5Y cells have also been recognized as a crucial factor in neuroprotection and neuroregeneration [56]. This was achieved through the increase in brain-derived neurotrophic factor (BDNF), which is well-known for enhancing neuronal survival, differentiation, and synaptic plasticity, along with its receptor TrkB [56]. However, further investigation is necessary to completely understand its mechanisms in neural regulation.

(The cited study is the only relevant study we could find.)